# LEARNING DYNAMIC STABILITY LANDSCAPES FROM GRAPH TOPOLOGY

## ABSTRACT

The robustness of synchronization is a central theme of the study of dynamical systems on networks. Typically one attempts to define a single stability index that characterizes the robustness of individual nodes to a class of perturbations. The dependence of a stability index on topology and system parameters can then be studied using network science or GNNs. Here we propose a novel upstream task, Stability Landscapes, that allows deriving many downstream stability indices. To support this task, we release two computationally intensive datasets of 10,000 graphs each at 20 and 100 nodes with per-node landscape labels. The dynamics are given by a conceptual oscillator model that captures aspects of the synchronization behavior of power grids. A compact graph neural network with a ~~lightweight~~ CNN decoder predicts these landscapes with about ~~90% $R^2$~~ 85% SSIM in distribution and ~~66% to 70%~~ 67% under a 20 to 100 size shift, and ~~60%-86% $R2$~~ 65%-73% SSIM when going from the 100 node ensemble to realistic power grid topologies with 100-400 nodes. This demonstrates that while basin landscapes are not suitable for study with conventional methods of network science, they are amenable to machine learning methods. This suggests that there is considerable potential in the study of complex networked systems across biology, neuroscience, and power grids, to move beyond scalar stability indices.

## 1 INTRODUCTION

Networks of coupled oscillators play a fundamental role in modeling both natural and engineered systems. They provide a unifying framework for understanding complex dynamics across fields, such as biology, neuroscience, ecology, physics, and engineering. Systems including the heart, the brain, food webs, coupled lasers, chemical reactions, power grids, and even firefly populations can all be described as oscillators interacting on complex networks Strogatz (2000); Acebron et al. (2000); Pikovsky et al. (2001); Acebrón et al. (2005); Pecora et al. (2014); Rodrigues et al. (2016). The behavior of these systems is strongly influenced by the underlying network topology which governs who interacts with whom and how perturbations spread.

An important phenomenon in oscillator networks is synchronization. Whether synchronization is desirable depends on the application. In the brain, excessive synchronization can signal dysfunction, as in epilepsy. In power grids, synchronization is essential for stable operation. This duality makes the robustness of the synchronous state a central question in networked complex systems. Understanding and controlling synchronization has practical implications that range from disrupting pathological synchronization in the brain Tass (2007) to designing infrastructure with favorable stability properties Yamamoto et al. (2023); Menck et al. (2013; 2014); Berner et al. (2021).

An essential question is thus whether the synchronous state is stable with respect to perturbations. Of special practical importance are local perturbations, that is, perturbations located at a specific node $i$ of the network. Given a dynamical system on a graph $\mathcal{G} = (\mathcal{V}, \mathcal{E}, \mathbf{X})$ with vertices $\mathcal{V}$, edges $\mathcal{E}$ and dynamical parameters $\mathcal{X}$, and a perturbation model with parameters $p$, we can then ask whether the synchronous state survives a perturbation at a node $i \in \mathcal{V}$:

$$s(\mathcal{G}, i, p) \in \{0, 1\}, \tag{1}$$

where $s = 1$ denotes survival. This function is extremely complex, and potentially expensive to evaluate numerically, as it requires long time-domain simulations of the whole graph. In the best case we can hope that the evaluation cost scales as $|\mathcal{E}|$. Further, it can depend very sensitively on $p$, as the region $s(\mathcal{G}, i, p)$ can have fractal boundaries or even more complex properties, such that every point in a region is also on its boundary **?**.

In practice the precise dependence on $p$ is not meaningful. Perturbation parameters are never known exactly. A central tool in the study of complex oscillator networks has thus become the probabilistic stability Menck et al. (2014); Hellmann et al. (2016). That is, we average over perturbation parameters $p$ using some measure $\rho$, to obtain a graph function

$$\mathrm{SN}_i^{(s)}(\mathcal{G}) = \int s(\mathcal{G}, i, p)\rho(p)dp \tag{2}$$

This function provides a node labeling of the graph $\mathcal{G}$. Every node $i$ is labeled by the probability $\mathrm{SN}_i^{(s)}$ that a perturbation randomly drawn from $p$ is survived. Estimating this function using Monte-Carlo integration to some accuracy requires a fixed number of evaluations of $s$ per node, labeling the whole graph thus scales as $|\mathcal{V}||\mathcal{E}|$ in the best case. For conceptual models of power grids, this labeling has become an important tool for understanding the impact of topological features on overall stability properties Witthaut et al. (2022); Kim et al. (2016); Nitzbon et al. (2017). Further it was shown in Nauck et al. (2022b;a; 2023) that these labels can be well predicted by GNN methods. Not only are these GNN evaluation dramatically faster than evaluating $\mathrm{SN}_i^{(s)}$ by sampling, they also scale as $|\mathcal{E}|$ with graph size.

A disadvantage of this approach is that it requires choosing a meaningful $\rho(p)$ when generating the dataset. Different situations of practical interest, and different stability aspects are captured by different $\rho$ or by more complex expectation values $\int O(p)s(G, i, p)\rho(p)dp$ of some observable $O$. This motivates us to introduce in this paper a new upstream task for stability prediction. We can decompose the space of perturbations into regions of similar parameters and average these together. Label a region by $h$, and introduce the measures $\rho(p \mid h)$ concentrated on the region $h$, then we can introduce the stability landscape for node $i$:

$$\mathrm{L}_i^{(s)}(\mathcal{G}, h) = \int s(\mathcal{G}, i, p)\rho(p \mid h)dp \tag{3}$$

From this, a number of downstream tasks can be evaluated easily. Intuitively, $\mathrm{L}_i^{(s)}(\mathcal{G}, \cdot)$ can be visualized as a stability landscape, that is, a heatmap over the perturbation space which highlights safe and unsafe regions for node $i$. Specifically, Figure 1 (a) shows a contour rendering of such a landscape for one node.

Most importantly, for $\rho(p) = \int \rho(p|h)\rho(h)dh$, then the scalar score is recovered as a mixture of regional probabilities:

$$\mathrm{SN}_i^{(s)}(\mathcal{G}) = \int \rho(h)\,\mathrm{L}_i^{(s)}(\mathcal{G}, h)\,dh. \tag{4}$$

The concrete case we will study in this paper, chooses $s = \mathrm{BS}$ as basin stability Menck et al. (2013), that is, perturbations are modeled as displacements of the variables associated to a node by a perturbation vector $p$, and are labeled as stable if the system returns to synchrony from this initial condition as time goes to infinity. This gives rise to the single node basin stability (SNBS) and the basin stability landscape (LBS), where Eq. (2) and Eq. (3) can be rewritten as:

$$\mathrm{SNBS}_i(\mathcal{G}) = \int \mathrm{BS}(\mathcal{G}, i, p)\rho(p)dp, \quad \text{and} \quad \mathrm{LBS}_i(\mathcal{G}, h) = \int \mathrm{BS}(\mathcal{G}, i, p)\rho(p \mid h)dp. \tag{5}$$

In this context, plotting $s(\mathcal{G}, i, p)$ as a function of $p$ is typically done for illustrative purposes Menck et al. (2014); Hellmann et al. (2020); Zhang et al. (2024).

As we are typically only interested in perturbations up to a maximum size, this leads naturally to a finite set of regions. Thus $\mathrm{LBS}_i(\mathcal{G}, h)$ provides a labeling of the graph where every node is labeled by a vector. In the concrete model we will study below, the perturbation vector is two-dimensional,

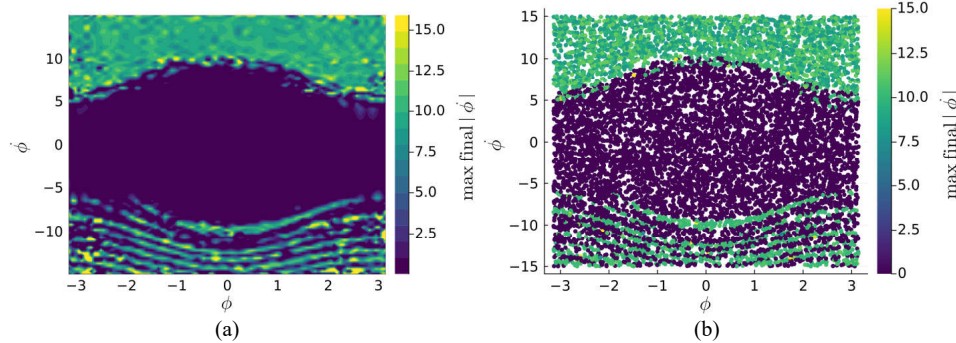

Figure 1: (a) Contour plot of the asymptotic deviation from synchrony, and (b) its Monte Carlo origin landscape for 10,000 perturbations at the same node $i$. The axes represent the perturbations in $p = \{\phi, \dot{\phi}\}$. The color encodes the maximum final frequency deviation from the set point, with darker colors indicating more stable regions.

and it is natural to discretize it into boxes, leading to cell-wise labels $\mathrm{LBS}_{i,m,n} \in [0, 1]$. In this work we extend the publicly available $\mathrm{SNBS}_i$ datasets for ensembles of oscillator networks Nauck et al. (2022a; 2023) by these basin landscapes.

This dataset includes two graph ensembles, 10,000 graphs with 20 nodes each and 10,000 graphs with 100 nodes each, to provide a valuable testbed for evaluating models' generalization capabilities. Specifically, the availability of two distinct graph scales allows researchers to effectively assess out-of-distribution (OOD) generalization performance across graph sizes, which is especially important given that labeling a full graph scales at least as $|\mathcal{V}||\mathcal{E}|$. Consequently, an ML model capable of generalizing from smaller (e.g., 20-node graph) to larger networks(e.g., 100-node graph) could provide the only practically viable route towards applying complex stability measures in practice.

The main contributions of this work are as follows:

- We introduce a novel stability assessment task, $L_i^{(s)}$ that captures advantages of probabilistic approaches, while also enabling varied downstream tasks. This is the first time that $L_i^{(s)}$ has been considered quantitatively.

- We present new datasets to support research on this task. The datasets consist of two ensembles, each containing 10,000 graphs with basin landscapes as prediction targets. They were generated using computationally expensive simulations requiring roughly **500,000** CPU hours.

- As an initial baseline, we propose a simple yet effective architecture combining a graph neural network (GNN) with a multilayer perceptron (MLP), demonstrating the potential of machine learning methods to address this critical challenge in networked dynamical systems.

- We establish the first practical benchmark for stability landscape prediction. Our models reach about 90% $R^2$ on heatmaps in distribution and 66% to 70% under a size shift from 20 to 100 nodes, transfer in a zero shot setting to four real power grids with up to 86.5% $R^2$ on heatmaps and up to 82.3% on SNBS, and match Monte Carlo fidelity while cutting evaluation from thousands of CPU hours across the datasets to seconds per graph.

To the best of our knowledge, the dataset we introduce possesses unique characteristics, as we are unaware of any similar dataset where images (landscapes) are predicted based on graph structures. This work has the potential to inspire other fields where similar image representations can be generated from graph topologies.

## 2 GENERATION OF DATASETS WITH BASIN LANDSCAPES

In this section, we present details on how to generate a dataset suitable for the novel supervised ML task. We begin by introducing the underlying dynamical system (the second-order Kuramoto

oscillator) and explain the methodology for generating basin stability landscapes via perturbations (see Section 2.1). We then outline how these basin landscapes are converted into heatmaps that are regarded as prediction targets for supervised ML (see Figure 1 (b)). Finally, we summarize the key statistical properties of our dataset (see Section 2.2).

## 2.1 BASIN LANDSCAPES OF DYNAMICAL SYSTEMS

The dynamical systems studied in this work are coupled networks of Kuramoto oscillators. We model each node of these networks as a paradigmatic second-order Kuramoto model Kuramoto (1984; 1975):

$$\ddot{\phi}_i = P_i - \alpha\dot{\phi}_i - \sum_{j=1}^{n} KA_{ij}\sin(\phi_i - \phi_j), \tag{6}$$

where $\phi_i$ is the phase angle at node $i$, $\dot{\phi}_i$, and $\ddot{\phi}_i$ are its first and second time derivatives, respectively. The network topology is encoded in the adjacency matrix $\mathbf{A}$. For all ensembles we fix the damping coefficient to $\alpha = 0.1$ and assume homogeneous coupling $K = 9$. The only node-dependent parameter is the injected power $P_i \in \{-1, +1\}$, which is regarded as the node feature in our ML settings. All dynamical simulations are conducted with respect to a reference frequency. Negative frequencies indicate that the frequency is below the reference frequency, e.g., 50 Hz in the case of the European power grid.

To perform the dynamical simulations, the statically stable state is perturbed using perturbations uniformly sampled in the phase-frequency space $(\phi, \dot{\phi}) \in [-\pi, \pi] \times [-15, 15]$. We apply 10,000 independent perturbations per node, with the results visualized as basin landscapes (see Figure 1 (b)). Each point represents the outcome of one dynamical simulation, and the plot shows the maximum absolute value of the final frequency at all nodes. Darker points (small values of *max final $|\dot{\phi}|$* ) indicate stable outcomes, whereas lighter points indicate unstable.

Since real applications usually care about whether a given combination of parameter configurations leads to stable or unstable states, rather than the exact value of $|\dot{\phi}|$, we classify simulation outcomes (i.e., points) as either stable or unstable. Following prior work (Nauck et al., 2022b), we adopt the threshold $|\dot{\phi}| \leq 0.1$, considering configurations below this value to be synchronized (dynamically stable).

On this basis, the landscape of each node is represented by 10,000 points, each point being labeled as either stable or unstable. The overall stability probability, known as single-node basin stability (SNBS) Menck et al. (2013), that a node returns to an overall stable state is defined as the ratio of stable points to the total number of points (i.e., the total number of perturbations applied). For the example in Figure 1 (b), this SNBS value equals the number of dark-purple points divided by the total of 10,000 perturbations applied to the node. Unlike previous univariate node-level regression-based studies (Nauck et al., 2022b;a; 2023), which predict only this scalar SNBS value for each node, we aim to reconstruct the full landscape directly from network topology.

Directly feeding raw basin landscapes into a learning algorithm is not feasible. Each node is perturbed $10,000$ times at randomly chosen phases (i.e., frequency pairs), resulting in a point cloud that differs from node to node in both location and density. Without a common "input shape", standard regression losses or neural network decoders cannot be applied.

Let a single perturbation be denoted by $\boldsymbol{\delta} = (\phi, \dot{\phi}) \in [-\pi, \pi] \times [-15, 15]$, and let the set of all perturbations applied to one node be $\mathcal{D} = \{\boldsymbol{\delta}_1, \ldots, \boldsymbol{\delta}_{10,000}\}$. To address this, we down-sample each landscape onto a regular $20 \times 20$ grid.

Denote by $\mathcal{C}_{i,m,n} \subset \mathcal{D}$ the subset of perturbations that fall into grid cell $(m, n)$ for the designated node $i$, and let $q_{i,m,n} = |\mathcal{C}_{i,m,n}|$. We define the empirical *stability probability* in that cell as

$$\text{LBS}_{i,m,n} = \frac{\#\{\boldsymbol{\delta} \in \mathcal{C}_{i,m,n} : |\dot{\phi}(\boldsymbol{\delta})| \leq 0.1\}}{q_{i,m,n}} \in [0, 1], \tag{7}$$

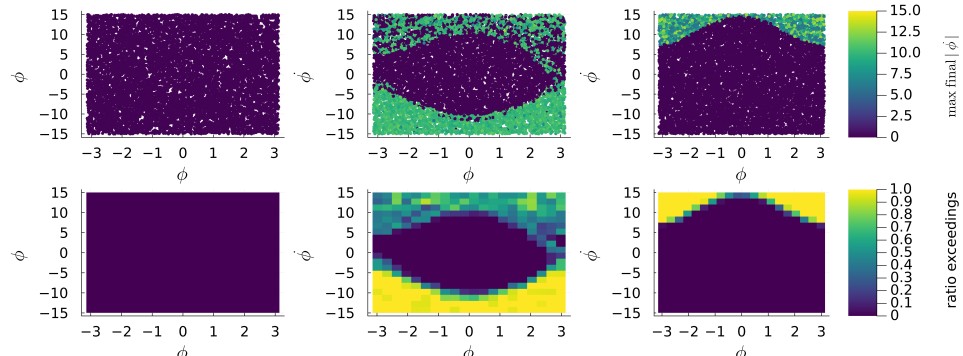

Figure 2: Examples of basin landscapes, original MC ~~sampleo~~ samples (top) and derived Basin Landscape with 20x20 grid cells (bottom).

Table 1: Dataset statistics and official splits (%). SNBS is reported as mean over all nodes.

| Type | Name | #Nodes / graph | #Total Edges | #Graphs | train/val/test | $\overline{\text{SNBS}}$ |
|---|---|---|---|---|---|---|
| Synthetic | dataset20 | 20 | 538,188 | 10,000 | 70/15/15 | 0.8374 |
| | dataset100 | 100 | 2,857,882 | 10,000 | 70/15/15 | 0.8735 |
| Realistic | Germany | 438 | 1,324 | 1 | Test only | 0.9045 |
| | France | 146 | 446 | 1 | Test only | 0.8757 |
| | GB | 120 | 165 | 1 | Test only | 0.8368 |
| | Spain | 98 | 350 | 1 | Test only | 0.9288 |

where we use the same threshold as in prior work. Cells with $\text{LBS}_{i,m,n} = 1$ are fully stable and cells with $\text{LBS}_{i,m,n} = 0$ are fully unstable. Stacking the $20 \times 20$ values yields $\mathbf{L}^{(i)} \in [0,1]^{20 \times 20}$, the per-node heatmap label. While the individual error of $\text{LBS}_{i,m,n}$ is quite large, the overall landscapes that emerge are quite robust, see Figure 2.

Let $\mathcal{N}_i = \sum_{m,n} q_{i,m,n}$ be the number of trajectories for node $i$ and define $w_{i,m,n} = q_{i,m,n}/\mathcal{N}_i$. By construction,

$$\text{SNBS}_i(\mathcal{G}) \;=\; \sum_{m,n} w_{i,m,n}\, \text{LBS}_{i,m,n}, \qquad \mathcal{N}_i = 10{,}000 \text{ in our dataset.} \tag{8}$$

This makes the landscape strictly richer than the classical scalar while remaining backward compatible. Figure 2 contrasts raw point clouds with their corresponding downsampled heatmaps. As shown in Theorem 3.1, reconstructing such a heatmap keeps the previous SNBS prediction error under control.

## 2.2 PROPERTIES OF THE DATASET

The dataset consists of two ensembles: dataset20 and dataset100. Each sample is a triple $(\mathcal{G}, i, \mathbf{L}^{(i)})$, where $\mathcal{G} = (\mathcal{V}, \mathcal{E}, \mathbf{X})$ is an undirected graph, $i \in \mathcal{V}$ is the designated node, and $\mathbf{L}^{(i)} \in [0,1]^{20 \times 20}$ is its basin-landscape heatmap label. $X_i = P_i \in \{+1, -1\}$ denotes the node feature, indicating power injection (see Equation (6)). Each ensemble is divided into training, validation, and test sets in a 70:15:15 ratio. Basic information about the datasets is summarized in Table 1. The histograms for the downstream task SNBS are shown in Figure 6 in Appendix B.1. Notably, for dataset100, there are more nodes that remain stable throughout. This difference in distribution poses a challenge for the out-of-distribution generalization task.

Furthermore, this dataset may help address the lack of benchmarks featuring extensive long-range dependencies, as the studied problem of dynamic stability inherently involves non-local behavior, and deeper GNNs demonstrate superior performance (Nauck et al., 2022a; 2023; 2024b;a). Combined with the task of image prediction, the dataset offers a challenging benchmark for developing GNN architectures capable of handling long-range dependencies. This is particularly relevant for

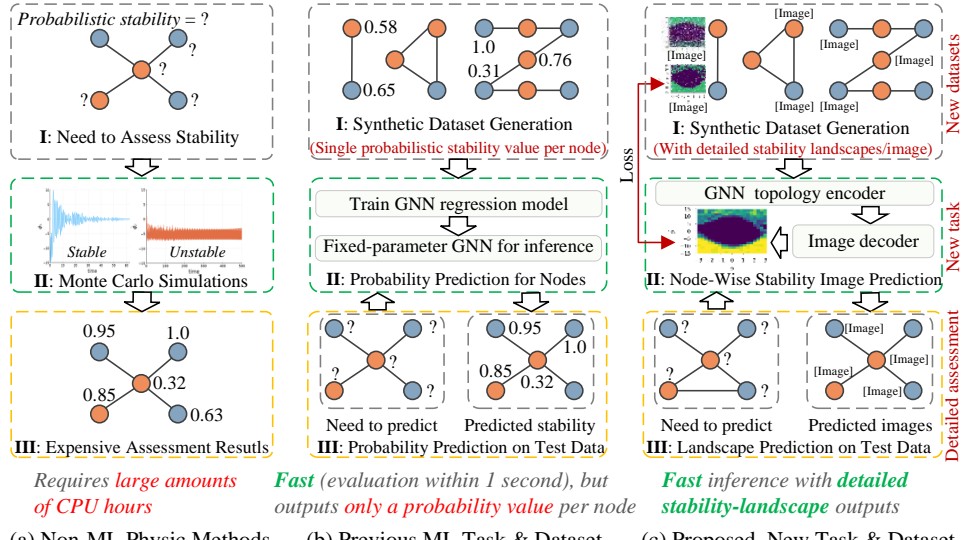

Figure 3: Comparison of stability-assessment workflows. (a) Traditional physics-based methods rely on costly Monte Carlo simulations to evaluate single-node basin stability, requiring large amounts of CPU hours. (b) Prior ML approaches (Nauck et al., 2022b;a; 2023) replace simulation with a trained GNN that predicts a probabilistic stability value per node in seconds, but sacrifices spatial detail. (c) In this work, we introduce a new dataset of full basin-landscape heatmaps and formulate a novel ML task: a GNN topology encoder plus image decoder jointly predict detailed, node-wise stability landscapes in one fast, end-to-end pass.

applications such as power grids, where long-range interactions play a critical role (Ringsquandl et al., 2021).

## 3 ARCHITECTURE: ENCODER-DECODER DESIGN AND OPTIMIZATION

We formulate basin landscape prediction as a supervised mapping:

$$f_\theta : (\mathcal{G}, i) \longrightarrow \mathbf{L}^{(i)} \in [0, 1]^{20 \times 20}, \tag{9}$$

where each graph $\mathcal{G} = (\mathcal{V}, \mathcal{E}, \mathbf{X})$ carries one binary node feature $X_i = P_i \in \{-1, 1\}$ and $\mathbf{L}^{(i)}$ is the heatmap of a designated node. Figure 3 (c) illustrates our architecture, which places the model within the broader workflow of stability assessment.

### 3.1 TOPOLOGY ENCODER AND IMAGE DECODER

**Topology encoder for embedding graph structural information.** The encoder $g_\phi$ is a message–passing graph neural network that converts the irregular topology into latent node embeddings $\mathbf{Z} = g_\phi(\mathcal{G}) \in \mathbb{R}^{|\mathcal{V}| \times d}$. Previous studies (Nauck et al., 2022a; 2023; 2024b;a) suggest that SNBS is not purely a local property but relies on long-range dependencies. The importance of such dependencies has also been independently observed in other power grid applications (Nauck et al., 2022a; 2023; 2024b;a). Given this evidence, GNNs with long-range capabilities are well-suited to serve as the encoder $g_\phi$. In this study, we utilize Topology Adaptive Graph convolution (TAG) (Du et al., 2017) and Dirac–Bianconi GNN (DBGNN) (Nauck et al., 2024a) convolution. TAG uses learnable polynomial filters to aggregate information from up to $K = 3$ hops per layer. DBGNN incorporates multiple micro-propagation steps within each layer, allowing the model to efficiently capture multi-hop information in a single layer.

**Decoder for reconstructing basin landscape.** For the chosen node we extract its embedding $z_i \in \mathbb{R}^d$ and pass it to a lightweight MLP $h_\psi : \mathbb{R}^d \to \mathbb{R}^{400}$ or a convolutional neural network (CNN) decoder. The output is reshaped into a single-channel $20 \times 20$ image. Although advanced

generative decoders, such as diffusion models and variational autoencoders, present promising avenues for future exploration, in this initial investigation we employ a simple per-node MLP or CNN to maintain model simplicity and support clear evaluation on our newly introduced task and dataset.

## 3.2 LEARNING OBJECTIVE

Ground-truth heatmaps contain continuous stability probabilities, and therefore training employs the mean-squared error (MSE):

$$\mathcal{L} = \big\| h_\psi(z_i) - \mathbf{L}^{(i)} \big\|_2^2, \quad \text{where } z_i = \big(g_\phi(\mathcal{G})\big)_i. \tag{10}$$

Since SNBS equals the weighted average of the heatmap entries, this loss upper-bounds the downstream SNBS error, which is detailed in Theorem 3.1.

**Theorem 3.1** (Pixel MSE upper-bounds SNBS error)**.** *Fix a node $i$. For each grid cell $(m, n)$, let $q_{i,m,n} \geq 0$ be the number of trajectories in that cell and set $\mathcal{N}_i = \sum_{m,n} q_{i,m,n}$ and $w_{i,m,n} = q_{i,m,n}/\mathcal{N}_i$. Let $LBS_{i,m,n}, \widehat{LBS}_{i,m,n} \in [0,1]$ be the true and predicted stability probabilities. Define*

$$\text{SNBS}_i = \sum_{m,n} w_{i,m,n} \, LBS_{i,m,n}, \qquad \widehat{\text{SNBS}}_i = \sum_{m,n} w_{i,m,n} \, \widehat{LBS}_{i,m,n}.$$

*Then*

$$\big(\widehat{\text{SNBS}}_i - \text{SNBS}_i\big)^2 \leq \sum_{m,n} w_{i,m,n} \big(\widehat{LBS}_{i,m,n} - LBS_{i,m,n}\big)^2. \tag{11}$$

*The right-hand side is exactly the weighted pixel-wise MSE employed as the loss in Equation* (10) *when sample counts per cell are unequal.*

The proof is in Appendix B.2. Theorem 3.1 guarantees that minimizing the weighted pixel-level MSE already keeps the SNBS error under control. Consequently we do not need to append a second loss term such as $(\widehat{\text{SNBS}}_i - \text{SNBS}_i)^2$ to the training objective. Omitting this extra term also removes the weighting factor that would otherwise be required to balance two losses, so optimization remains a single-objective problem with one fewer hyper-parameter to tune. Moreover, the inequality provides a quantitative error guarantee. Even if the model makes large mistakes on a small subset of pixels, the global SNBS error cannot exceed the reported weighted-MSE. This gives reviewers and practitioners an interpretable upper bound: as long as the overall heatmap MSE is small, the classical stability score is automatically protected.

## 4 PREDICTIVE PERFORMANCE

The predictive performance of the model is assessed using two approaches. First, we evaluate the model's ability to predict heatmaps both qualitatively and quantitatively. Second, we assess its performance on a downstream task by predicting the volume of the basin stability, a metric derived from the heatmaps. Third, zero-shot transfer to four real grid topologies, namely Germany, France, Great Britain, and Spain. Unless noted otherwise, zero-shot uses the model trained on *dataset100*. All evaluations report ~~$R^2$ on the heatmaps and on~~ metrics on comparing the heatmaps, as well as the previously published task of SNBS derived from the heatmaps through Eq. (5).

A brief hyperparameter study was conducted to tune the model and training parameters. The DBGNN uses two layers, each with 10 internal propagation steps (alternating between node-to-line and line-to-node updates), effectively capturing information from nodes up to 10 edges away. For TAG, the parameter ( $K$=3 ) is selected with 5 layers, allowing the TAG model to consider nodes within 15 edge steps. Further information on the models and the training settings is provided in Appendix B.3.

### 4.1 EVALUATION: HEATMAP COMPARISON

The qualitative performance of the ML model is assessed by comparing the predicted heatmaps to the ground truth. Example heatmaps from the TAG-MLP model, trained and evaluated on dataset20, are shown in Figure 4. Additional results for TAG-MLP trained and evaluated on dataset100, as

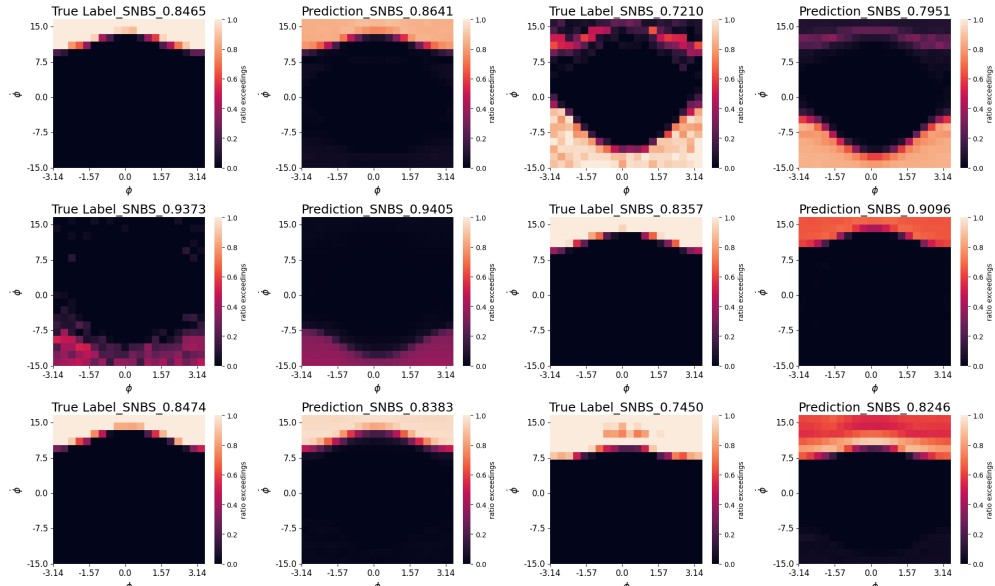

Figure 4: Comparison of true labels and predicted heatmaps for the the TAG-MLP model trained and evaluated on dataset20.

Table 2: Performance on predicting the heatmaps of the landscapes measured by $SSIM$ in %.

| Model | In-Distribution | | Out-of-Distribution |
|---|---|---|---|
| | tr20ev20 | tr100ev100 | tr20ev100 |
| TAG-MLP | $81.32_{\pm 0.55}$ | $81.61_{\pm 0.15}$ | $71.50_{\pm 0.28}$ |
| DBGNN-MLP | $80.63_{\pm 0.21}$ | $82.87_{\pm 0.79}$ | $74.03_{\pm 0.46}$ |
| TAG-CNN | $76.47_{\pm 0.37}$ | $70.51_{\pm 0.48}$ | $67.46_{\pm 0.56}$ |
| DBGNN-CNN | $\mathbf{84.24}_{\pm 0.27}$ | $\mathbf{85.89}_{\pm 0.21}$ | $\mathbf{78.50}_{\pm 0.19}$ |

well as for the out-of-distribution (OOD) task, trained on dataset20 and evaluated on dataset100, are provided in the appendix (Appendix B.4). Furthermore, we demonstrate the models' generalization capabilities on the real-world grid topologies of France, Germany, Great Britain, and Spain, as discussed in Section 4.3.

Overall, the visualizations demonstrate that the general shapes of the landscapes are often predicted accurately. However, there are instances where the predicted SNBS deviates significantly from the true labels, highlighting areas for improvement. These discrepancies can occur even when the qualitative comparison shows that the stable region's structure is correctly predicted, but the ratio of exceedances is not accurately captured. Thus, even if the predicted SNBS differs from the true value, the prediction may still be valuable as it correctly identifies the stable region.

The quantitative performance, summarized in ~~Table 8 using the coefficient of determination ($R^2$~~ Table 2 using the tructural similarity index measure (SSIM), demonstrates that the heatmaps can be predicted with an ~~$R^2$~~ SSIM of up to ~~90~~86%. Importantly, the approach also performs well in out-of-distribution generalization, albeit with noticeably lower performance. In general image decoder techniques such as convolutional neural network (CNN) outperform MLPs. Additional metrics are provided in Appendix B.5.

## 4.2 EVALUATION ON THE DOWNSTREAM TASK AT PREDICTING SNBS

As a second benchmark, we evaluate the downstream task of predicting SNBS based on these heatmap predictions. Similar to Nauck et al. (2023), we report the mean value of the best 3 seeds out of 5 initializations and the corresponding standard deviation.

The performance of the downstream task is presented in Table 3. The overall performance is comparable to the results achieved when directly predicting SNBS, bypassing the intermediate step of

Table 3: Performance on the downstream task of predicting SNBS based on heatmap predictions measured by $R^2$ in %.

| Model | In-Distribution | | Out-of-Distribution |
|---|---|---|---|
| | tr20ev20 | tr100ev100 | tr20ev100 |
| ArmaNet | $82.22_{\pm 0.12}$ | $88.35_{\pm 0.12}$ | $67.12_{\pm 0.80}$ |
| GCNNet | $70.74_{\pm 0.15}$ | $75.19_{\pm 0.14}$ | $58.24_{\pm 0.47}$ |
| TAGNet | $82.50_{\pm 0.36}$ | $88.32_{\pm 0.10}$ | $66.32_{\pm 0.74}$ |
| DBGNN | $\mathbf{85.68}_{\pm 0.10}$ | $\mathbf{90.08}_{\pm 0.02}$ | $\mathbf{73.73}_{\pm 0.07}$ |
| TAG-MLP | $82.77_{\pm 0.33}$ | $87.13_{\pm 0.30}$ | $62.93_{\pm 1.35}$ |
| DBGNN-MLP | $84.48_{\pm 0.35}$ | $89.69_{\pm 0.15}$ | $72.12_{\pm 0.66}$ |
| TAG-CNN | $66.22_{\pm 0.11}$ | $71.46_{\pm 0.13}$ | $48.89_{\pm 0.78}$ |
| DBGNN-CNN | $83.77_{\pm 0.25}$ | $89.32_{\pm 0.09}$ | $69.42_{\pm 0.87}$ |

Table 4: Out-of-distribution Performance on predicting the heatmaps of real topologies measured by SSIM in %.

| Model | tr100evGermany | tr100evFrance | tr100evGB | tr100evSpain |
|---|---|---|---|---|
| TAG-MLP | $\mathbf{80.82}_{\pm 0.68}$ | $\mathbf{77.52}_{\pm 0.31}$ | $\mathbf{83.49}_{\pm 0.53}$ | $\mathbf{76.59}_{\pm 0.45}$ |
| DBGNN-MLP | $47.56_{\pm 6.90}$ | $50.68_{\pm 5.90}$ | $44.20_{\pm 5.57}$ | $53.77_{\pm 6.09}$ |
| TAG-CNN | $66.77_{\pm 0.51}$ | $68.23_{\pm 0.62}$ | $72.88_{\pm 0.34}$ | $64.80_{\pm 0.49}$ |
| DBGNN-CNN | $32.88_{\pm 9.88}$ | $32.18_{\pm 7.89}$ | $30.02_{\pm 8.23}$ | $40.16_{\pm 13.27}$ |

predicting the landscapes. The slightly lower performance observed may stem from the increased complexity of accurately predicting the landscapes. Examining exemplary basin landscapes in Figures 4 and 8, we often observe symmetric structures with comparable probabilities. By symmetric structures, we refer to regions where stability is present either at low $\phi$ or high $\phi$, with only a small region in between where instabilities occur. When predicting only the probability, it suffices to approximate the size of this unstable region. However, predicting the landscapes requires not only estimating the size, but also accurately locating the unstable region. In contrast, for probability prediction, the specific location of the stable regions (whether at low or high $\phi$) is irrelevant. This added requirement of spatial precision in landscape prediction likely contributes to the observed reduction in overall performance. To conclude, the performance reduction is minimal, and the results serve as a proof of concept, highlighting the potential of using ML to predict landscapes.

### 4.3 GENERALIZATION TO REAL-WORLD TOPOLOGIES

To further evaluate generalization, we assess model performance on the real-world grid topologies of France, Germany, Great Britain, and Spain. As shown in ~~?? and table 5~~ Tables 4 and 5, the ML models generally adapt well to these realistic networks across both tasks. TAG-MLP, in particular, exhibits robust generalization to grids with hundreds of nodes, despite being trained solely on small synthetic graphs, suggesting strong scalability. The low downstream performance of DBGNN-MLP and the CNN decoders for SNBS prediction likely reflects the unique dynamical properties of the Great Britain grid (Nauck et al., 2024b), and underscores the complexity of the task.

Table 5: Out-of-distribution Performance on the downstream task of predicting SNBS measured by $R^2$ in %.

| Model | tr100evGermany | tr100evFrance | tr100evGB | tr100evSpain |
|---|---|---|---|---|
| TAG-MLP | $70.09_{\pm 0.68}$ | $\mathbf{82.30}_{\pm 0.15}$ | $\mathbf{59.46}_{\pm 3.03}$ | $\mathbf{68.08}_{\pm 0.92}$ |
| DBGNN-MLP | $16.60_{\pm 27.87}$ | $37.78_{\pm 23.4}$ | $-105.62_{\pm 63.05}$ | $8.41_{\pm 35.82}$ |
| TAG-CNN | $\mathbf{76.67}_{\pm 0.86}$ | $65.12_{\pm 2.08}$ | $-18.40_{\pm 3.53}$ | $58.63_{\pm 1.61}$ |
| DBGNN-CNN | $13.24_{\pm 21.56}$ | $35.02_{\pm 8.19}$ | $-64.96_{\pm 6.27}$ | $12.22_{\pm 13.23}$ |

### 4.4 IDENTIFICATION OF CRITICAL CONTINGENCIES

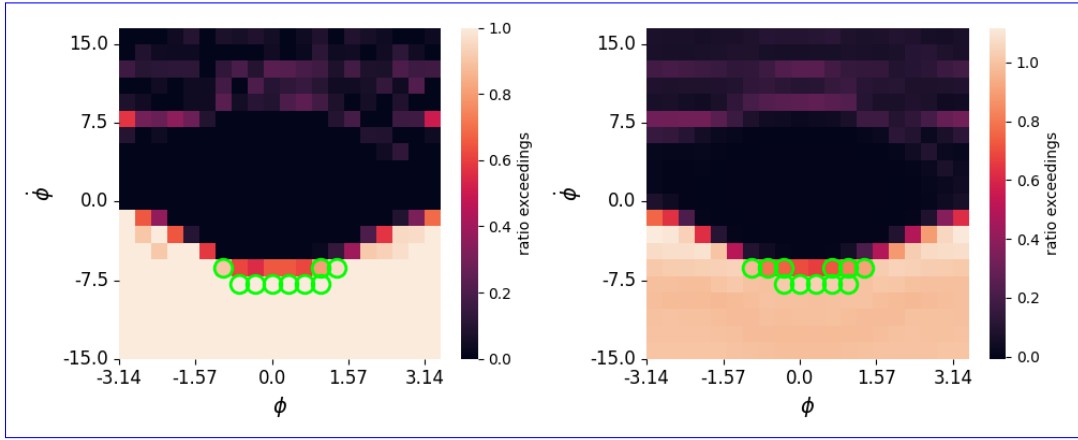

Figure 5: Visualization of critical contingencies (true labels on the left and predictions on the right).

The generated heatmaps enable a more detailed analysis for contingency screening, a key task for grid operators. By leveraging the stability landscape, it becomes straightforward to identify regions where minimal perturbations are most likely to destabilize the grid. These regions can then be prioritized by operators for further in-depth analysis to understand the underlying causes of system vulnerability—an approach that would be challenging to implement directly.

A common strategy involves analyzing the geometric features of the landscape space from which the system can recover to normal operation. This is a central topic in engineering and mechanics Ma et al. (2019), with ongoing research in the area Zhang & Strogatz (2021). A typical focus is on determining the size of the smallest disturbance capable of destabilizing the system Halekotte & Feudel (2020). The probabilistic analogue—namely, the radius at which the probability of desynchronization exceeds a certain threshold—is studied as the linear size of the basin in Delabays et al. (2017). The radial projection of our stability landscape is exactly what they estimate directly numerically. This could be an immediate downstream task using the predicted landscapes. From a power grid perspective, a further refinement is more interesting: Finding not just the radius, but also the precise direction of perturbation at which we first see a high likelihood of failure. In the power grid operator setting this provides a type of contingency screening: As not all possible contingencies can be studied, the basin landscape allows identifying a set of minimal perturbation regions that have a high likelihood of destabilizing the grid. These can then be singled out for further in-depth analysis, to understand why the system is prone to failure here.

To identify the most critical perturbations, we use the following procedure: For all pixels in the grid ($20 \times 20 \times N_{\text{nodes}}$), we select those with an exceeding ratio greater than 0.7. Among these, we identify the 20 most critical cells by lexicographic sorting—first by distance to the center, then by exceeding ratio. Quantitavive assessment is conducted in Appendix B.6. To illustrate the method, the most critical cells (predicted and true labels) are shown by lime circles for the most critical node of an example grid in Figure 5. Even though the critical node has a somewhat large overall stability (SNBS $\approx 0.6$) the contingency screening clearly identifies the smallest critical perturbations.

## 5 CONCLUSION

We introduce graph-to-field prediction of per-node stability landscapes from topology, replacing a single SNBS score with a field that localizes where failures arise. We release two large-scale datasets with landscape labels at 20 and 100 nodes that support in-distribution testing and size-shift evaluation. We present a compact GNN with a lightweight decoder and prove that a pixel loss upper bounds SNBS error, which turns training into a single objective with a clear guarantee. Our models match the fidelity of Monte Carlo while reducing evaluation from thousands of CPU hours across the datasets to seconds per graph. They generalize without retraining to four real power grid topologies. These pieces could establish a practical benchmark and a template for learning stability landscapes on graphs, and they invite landscape-level prediction across ~~biology~~other domains. We expect the framework to transfer to other dynamical systems ( biochemical, neuroscience, and infrastructure) given sufficient high-resolution training data.

## REPRODUCIBILITY STATEMENT

All code for dataset generation, model training, and analysis will be made publicly available on GitHub and Zenodo upon publication. Ready-to-use datasets for machine learning training will be hosted on HuggingFace.

To facilitate review and reproducibility, sample code for training a GNN on the novel dataset is available at `https://drive.google.com/drive/folders/1blfk9gjPT9extf2GY0occAzrtgOJdbpX?usp=drive_link`.

The repository includes tools for training, evaluation, hyperparameter optimization, and visualization, with instructions for environment setup and experiment replication.

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

# A APPENDIX

# B TECHNICAL APPENDICES AND SUPPLEMENTARY MATERIAL

The appendix is organized into four sections. The first section provides additional information on the datasets. The second section presents the theoretical proof for the theorem concerning the relationship between the heatmaps and SNBS. Next, details regarding the models and training procedures are described. Finally, the appendix concludes with supplementary results.

## B.1 FURTHER PROPERTIES OF THE DATASET

Figure 6 shows the histograms for the two ensembles and the downstream task target SNBS. The ensemble with graphs of size 100 contains more nodes that remain stable under all perturbations. This difference in distribution poses a challenge for machine learning models to generalize across ensembles of varying sizes.

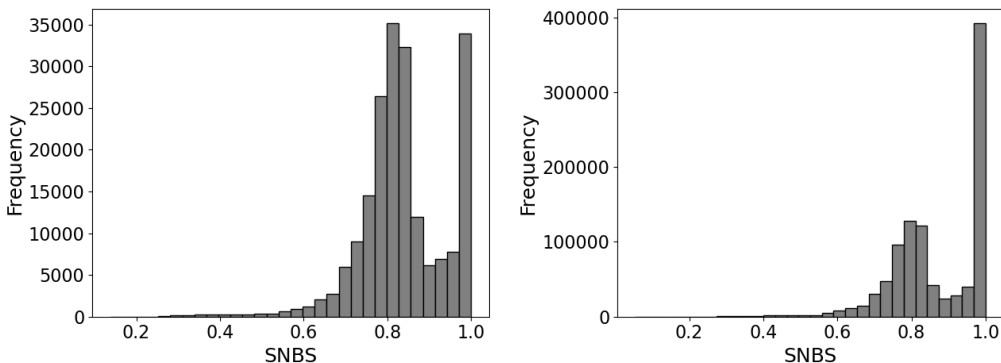

Figure 6: Histograms of the downstream task SNBS. The left panel corresponds to dataset20, and the right panel corresponds to dataset100.

## B.2 PROOF OF THEOREM 3.1.

The following presents the proof of Theorem 3.1.

**Theorem.** *Fix a node $i$. For each grid cell $(m, n)$, let $q_{i,m,n} \geq 0$ be the number of trajectories in that cell and set $\mathcal{N}_i = \sum_{m,n} q_{i,m,n}$ and $w_{i,m,n} = q_{i,m,n}/\mathcal{N}_i$. Let $LBS_{i,m,n}, \widehat{LBS}_{i,m,n} \in [0, 1]$ be the true and predicted stability probabilities. Define*

$$\mathrm{SNBS}_i = \sum_{m,n} w_{i,m,n} \, LBS_{i,m,n}, \qquad \widehat{\mathrm{SNBS}}_i = \sum_{m,n} w_{i,m,n} \, \widehat{LBS}_{i,m,n}.$$

*Then*

$$\left(\widehat{\mathrm{SNBS}}_i - \mathrm{SNBS}_i\right)^2 \leq \sum_{m,n} w_{i,m,n} \left(\widehat{LBS}_{i,m,n} - LBS_{i,m,n}\right)^2. \tag{12}$$

*The right-hand side is exactly the weighted pixel-wise MSE employed as the loss in Equation (10) when sample counts per cell are unequal.*

*Proof.* Let $\varepsilon_{i,m,n} = \widehat{\text{LBS}}_{i,m,n} - \text{LBS}_{i,m,n}$ and recall that the weights $w_{i,m,n} \geq 0$ satisfy $\sum_{m,n} w_{i,m,n} = 1$. By the definitions of (weighted) SNBS we have:

$$\left(\widehat{\text{SNBS}}_i - \text{SNBS}_i\right)^2 = \left(\sum_{m,n} w_{i,m,n} \widehat{\text{LBS}}_{i,m,n} - \sum_{m,n} w_{i,m,n} \text{LBS}_{i,m,n}\right)^2 \tag{13}$$

$$= \left(\sum_{m,n} w_{i,m,n}\, \epsilon_{i,m,n}\right)^2 \tag{14}$$

$$\leq \sum_{m,n} w_{i,m,n}\, \epsilon_{i,m,n}^2 \quad \text{(by Jensen's inequality)} \tag{15}$$

$$= \sum_{m,n} w_{i,m,n} \left(\widehat{\text{LBS}}_{i,m,n} - \text{LBS}_{i,m,n}\right)^2 \tag{16}$$

$$= \sum_{m=1}^{20} \sum_{n=1}^{20} w_{i,m,n} \left(\widehat{\text{LBS}}_{i,m,n} - \text{LBS}_{i,m,n}\right)^2, \tag{17}$$

which completes the proof. $\square$

### B.3 MODEL AND TRAINING PROPERTIES

Details of the final model architecture are summarized in Table 6, while the training properties are listed in Table 7. The CNN decoder replaces the MLP head with a three-layer transposed-convolutional decoder (channels $64 \rightarrow 32 \rightarrow 16$, single output channel, no dropout), while the GNN encoders remain unchanged. Hyperparameter optimization was performed using Optuna (Akiba et al., 2019). Training was conducted on a local HPC equipped with an Nvidia A100 GPU (80GB memory). For DBGNN-MLP, training on dataset20 with 5 consecutive seeds required approximately 21 hours and 35 minutes, while training on dataset100 took about 6 days. For TAG-MLP, training on dataset20 took less than 12 hours, and on dataset100 less than 2 days.

Table 6: Summary of model properties.

| Property | TAG-MLP | DBGNN-MLP |
|---|---|---|
| Number of GNN layers | 5 | 2 |
| Hidden dimension of GNN | 1751 | 512 |
| Output dimension of GNN | 671 | 512 |
| Hidden dimension of MLP | 654 | 512 |
| Output dimension of MLP | 512 | 512 |
| Dropout of GNN layer | $\approx 0.148$ | 0.04 |
| Dropout of MLP layer | 0.26 | 0.02 |

### B.4 ADDITIONAL RESULTS

Example heatmaps for TAG-MLP trained and evaluated on dataset100 are shown in Figure 7, while heatmaps for TAG-MLP trained on dataset20 and evaluated on dataset100 are presented in Figure 8. Consistent with the observations from Figure 4, the predictions capture most of the underlying structure accurately.

### B.5 ADDITIONAL RESULTS ON QUANTITATIVE COMPARISON OF HEATMAPS

In addition to Table 2, we also provide the performances measured by $R^2$ in Table 8 and Learned Perceptual Image Patch Similarity (LPIPS) Table 9.

In Tables 10 and 11, the out-of-distribution performances of the generated heatmaps are shown. The results confirm the overall observations from $SSIM$ from Table 4.

### B.6 DOWNSTREAM TASK: IDENTIFICATION OF CONTINGENCIES

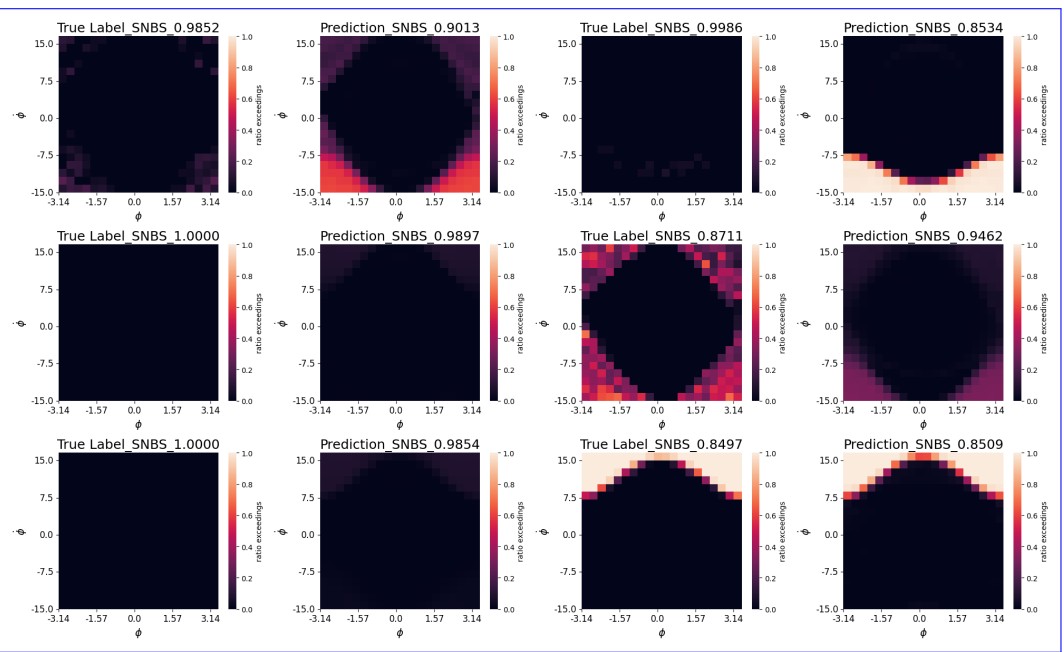

Figure 7: Comparison of true labels and predicted heatmaps for the the TAG-MLP model trained and evaluated on dataset100.

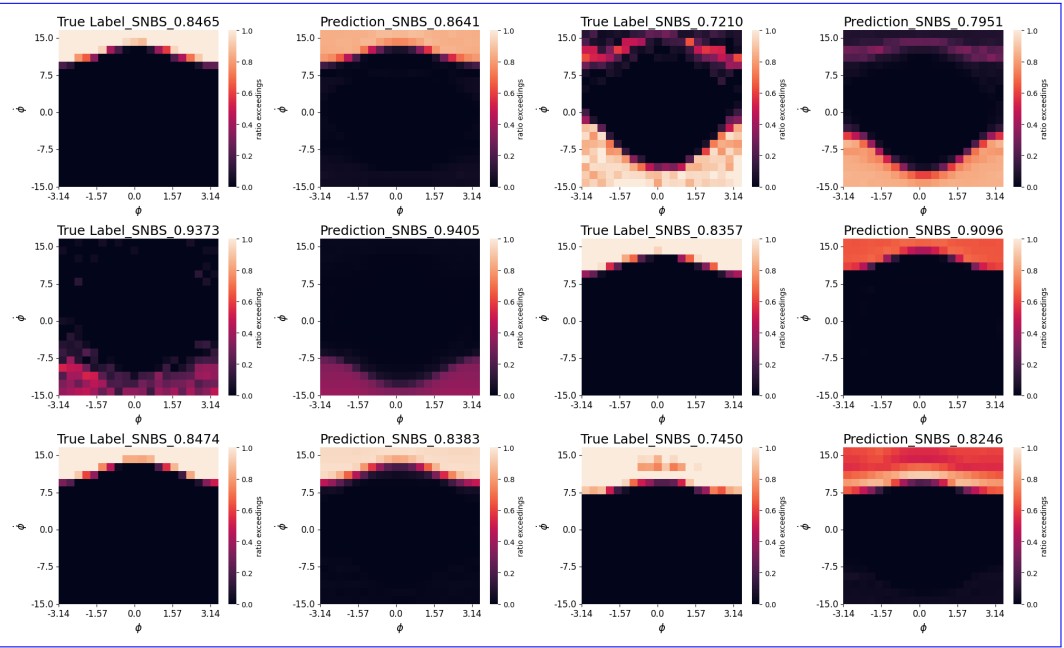

Figure 8: Comparison of true labels and predicted heatmaps for the the TAG-MLP model trained on dataset20 and evaluated on dataset100.

Table 7: Overview of training properties. MSE denotes mean squared error. The TAG–CNN and DBGNN–CNN decoders (trained on *dataset20* and *dataset100*) use the hyperparameters shown in the last column.

| Parameters | TAG-MLP (tr20) | TAG-MLP (tr100) | DBGNN-MLP (tr20) | DBGNN-MLP (tr100) | CNN decoders |
|---|---|---|---|---|---|
| Learning rate | $\approx$ 7.77e-05 | $\approx$ 6.77e-05 | $\approx$ 6.77e-06 | $\approx$ 6.77e-05 | $\approx$ 6.77e-05 |
| Number of epochs | 250 | 250 | 1000 | 1000 | 250 |
| Loss function | MSE | MSE | MSE | MSE | MSE |
| Training batch size | 32 | 32 | 32 | 32 | 32 |

Table 8: Performance on predicting the heatmaps of the landscapes measured by $R^2$ in %.

| Model | In-Distribution | | Out-of-Distribution |
|---|---|---|---|
| | tr20ev20 | tr100ev100 | tr20ev100 |
| TAG-MLP | $89.83_{\pm 0.11}$ | $88.86_{\pm 0.14}$ | $69.53_{\pm 0.53}$ |
| DBGNN-MLP | $\mathbf{90.44}_{\pm 0.18}$ | $\mathbf{90.61}_{\pm 0.13}$ | $\mathbf{76.50}_{\pm 0.31}$ |
| TAG-CNN | $83.21_{\pm 0.03}$ | $78.60_{\pm 0.04}$ | $61.73_{\pm 0.51}$ |
| DBGNN-CNN | $90.31_{\pm 0.10}$ | $90.38_{\pm 0.06}$ | $74.08_{\pm 0.83}$ |

Figure 5 visualized the most critical node in the first grid of dataset20. By also showing the second node, where critical cells are identified in Figure 9 all of the most 20 critical cells are visualized, since they only occurred at those two nodes.

Quantitative assessment is challenging because we aim to predict individual cells (pixels) rather than broader regions. For instance, predicting a neighboring cell may be acceptable if it has similar stability, but it can be problematic if the neighboring cell is actually stable. Therefore, we use a strict evaluation procedure that only considers a prediction correct if the exact same cell is identified.

Predictive performance is reported in Tables 12 and 13 using the intersection over union metric, which quantifies the overlap between the 20 most critical cells in the predicted and true heatmaps.

## B.7 INVESTIGATING THE IMPACT OF GRID RESOLUTION

To evaluate the effect of grid resolution on the analysis, we test various cell densities beyond the standard $20 \times 20$ configuration. Figure 10 shows heatmaps generated with three different resolutions: 5, 10, 20, and 30 cells per axis. The $20 \times 20$ grid represents an good balance between capturing fine spatial structures and maintaining sufficient sample density within each cell for statistical reliability.

The performances in Tables 14 to 16 and Table 17 confirm that the approach is feasible with different grid sizes. However, the performance with grids of size $10 \times 10$ is lower in comparison to $20 \times 20$.

Table 9: Performance on predicting the heatmaps of the landscapes measured by $LPIPS$ in %.

| Model | In-Distribution | | Out-of-Distribution |
|---|---|---|---|
| | tr20ev20 | tr100ev100 | tr20ev100 |
| TAG-MLP | $16.47_{\pm 0.25}$ | $14.72_{\pm 0.15}$ | $22.23_{\pm 0.02}$ |
| DBGNN-MLP | $17.33_{\pm 0.60}$ | $13.71_{\pm 0.31}$ | $20.61_{\pm 0.52}$ |
| TAG-CNN | $19.89_{\pm 0.18}$ | $24.19_{\pm 0.36}$ | $25.83_{\pm 0.31}$ |
| DBGNN-CNN | $\mathbf{13.71}_{\pm 0.18}$ | $\mathbf{12.35}_{\pm 0.21}$ | $\mathbf{17.10}_{\pm 0.12}$ |

Table 10: Out-of-distribution Performance on predicting the heatmaps of real topologies measured by $R^2$ in %.

| Model | tr100evGermany | tr100evFrance | tr100evGB | tr100evSpain |
|---|---|---|---|---|
| TAG-MLP | $60.66_{\pm 1.27}$ | $\mathbf{82.64}_{\pm 0.39}$ | $\mathbf{86.50}_{\pm 0.80}$ | $\mathbf{68.50}_{\pm 1.00}$ |
| DBGNN-MLP | $31.06_{\pm 19.80}$ | $53.68_{\pm 10.32}$ | $51.08_{\pm 9.33}$ | $32.88_{\pm 13.50}$ |
| TAG-CNN | $\mathbf{73.15}_{\pm 0.72}$ | $72.95_{\pm 1.14}$ | $63.68_{\pm 0.89}$ | $66.60_{\pm 0.85}$ |
| DBGNN-CNN | $37.89_{\pm 3.32}$ | $57.76_{\pm 2.56}$ | $60.56_{\pm 2.00}$ | $40.08_{\pm 7.04}$ |

Table 11: Out-of-distribution Performance on predicting the heatmaps of real topologies measured by $LPIPS$ in %.

| Model | tr100evGermany | tr100evFrance | tr100evGB | tr100evSpain |
|---|---|---|---|---|
| TAG-MLP | $\mathbf{13.64}_{\pm 0.26}$ | $\mathbf{17.44}_{\pm 0.55}$ | $\mathbf{14.44}_{\pm 0.17}$ | $\mathbf{15.90}_{\pm 0.21}$ |
| DBGNN-MLP | $34.35_{\pm 3.89}$ | $32.01_{\pm 2.82}$ | $33.09_{\pm 2.36}$ | $30.40_{\pm 3.44}$ |
| TAG-CNN | $26.18_{\pm 0.39}$ | $25.02_{\pm 0.48}$ | $19.81_{\pm 0.28}$ | $27.58_{\pm 0.30}$ |
| DBGNN-CNN | $42.28_{\pm 5.12}$ | $42.30_{\pm 3.95}$ | $42.91_{\pm 4.80}$ | $39.06_{\pm 7.13}$ |

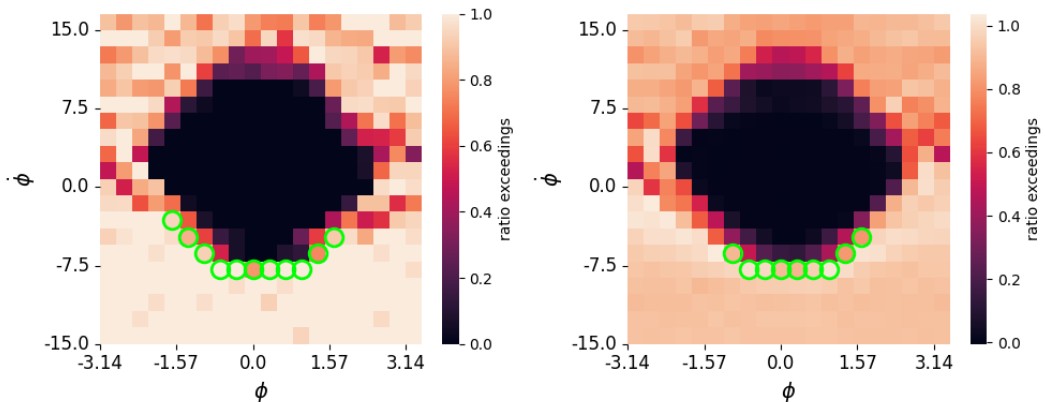

Figure 9: Visualization of critical contingencies for node 16 in the first grid of the test set from dataset20. True labels are shown on the left, and predictions are shown on the right.

Table 12: Performance on the downstream task of identifying critical contingencies, measured by intersection over union (%) between predicted and true critical cells.

| Model | In-Distribution | | Out-of-Distribution |
|---|---|---|---|
| | tr20ev20 | tr100ev100 | tr20ev100 |
| TAG-MLP | $0.58_{\pm 0.01}$ | $0.41_{\pm 0.019}$ | $0.32_{\pm 0.018}$ |
| DBGNN-MLP | $0.61_{\pm 0.03}$ | $0.51_{\pm 0.01}$ | $0.41_{\pm 0.04}$ |

Table 13: Out-of-distribution performance on the downstream task of identifying critical contingencies, measured by intersection over union (%) between predicted and true critical cells.

| Model | tr100evGermany | tr100evFrance | tr100evGB | tr100evSpain |
|---|---|---|---|---|
| TAG-MLP | $57.5_{\pm 0.03}$ | $20.7_{\pm 0.06}$ | $26.6_{\pm 0.02}$ | $6.0_{\pm 0.05}$ |
| DBGNN-MLP | $17.9_{\pm 0.20}$ | $0.0_{\pm 0.0}$ | $5.4_{\pm 0.04}$ | $3.1_{\pm 0.02}$ |

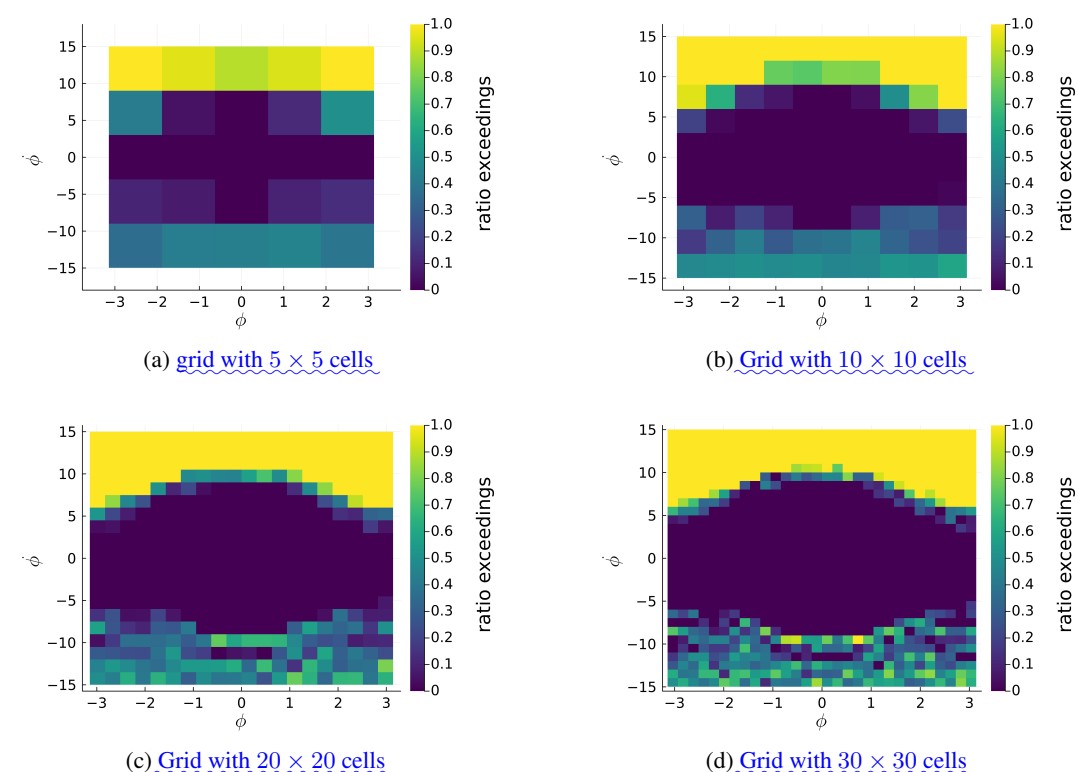

(a) grid with $5 \times 5$ cells

(b) Grid with $10 \times 10$ cells

(c) Grid with $20 \times 20$ cells

(d) Grid with $30 \times 30$ cells

Figure 10: Comparison of the grid size of the heatmaps with the following number per axis: 5, 10, 20, 30.

Table 14: Performance on predicting the heatmaps of the landscapes measured by SSIM in % with a grid size of $10 \times 10$.

| Model | In-Distribution | | Out-of-Distribution |
|---|---|---|---|
| | tr20ev20 | tr100ev100 | tr20ev100 |
| TAG-MLP (ns10) | $81.88_{\pm 0.39}$ | $82.45_{\pm 0.77}$ | $67.68_{\pm 0.02}$ |

Table 15: Performance on predicting the heatmaps of the landscapes measured by LPIPS in % with a grid size of $10 \times 10$.

| Model | In-Distribution | | Out-of-Distribution |
|---|---|---|---|
| | tr20ev20 | tr100ev100 | tr20ev100 |
| TAG-MLP (ns10) | $10.04_{\pm 0.21}$ | $9.63_{\pm 0.28}$ | $17.18_{\pm 0.40}$ |

Table 16: Performance on predicting the heatmaps of the landscapes measured by $R^2$ in % with a grid size of $10 \times 10$. For a grid size of $20 \times 20$, the results are shown in Table 8.

| Model | In-Distribution | | Out-of-Distribution |
|---|---|---|---|
| | tr20ev20 | tr100ev100 | tr20ev100 |
| TAG-MLP (ns10) | $90.63_{\pm 0.15}$ | $89.21_{\pm 0.29}$ | $70.32_{\pm 0.06}$ |

Table 17: Performance on the downstream task of predicting SNBS based on heatmap predictions measured by $R^2$ in % with a grid size of $10 \times 10$. For a grid size of $20 \times 20$, the results are shown in Table 3.

| Model | In-Distribution | | Out-of-Distribution |
|---|---|---|---|
| | tr20ev20 | tr100ev100 | tr20ev100 |
| TAG-MLP (ns10) | $82.15_{\pm 0.36}$ | $86.28_{\pm 0.43}$ | $62.98_{\pm 1.10}$ |

