# OpenReview forum: "Learning Dynamic Stability Landscapes from Graph Topology"
_ICLR.cc/2026/Conference — Submitted to ICLR 2026_

### Official Review · Reviewer_yScr · 2025-10-31

**Soundness:** 4
**Presentation:** 4
**Contribution:** 3
**Rating:** 8
**Confidence:** 4

**Summary:**

The authors lay out a full, novel computational pipeline, from conceptual premises to data set generation to a first baseline model, for the problem of predicting the behavior of coupled oscillators under perturbation. They focus on the interesting case of understanding how individual nodes behave qualitatively (whether or not they remain synchronized with respect to a rotating reference frame) under the influence of perturbations to their phase and frequency. Instead of predicting the effect of a single perturbation, they investigate the more general problem of inferring an entire landscape of perturbation effects for all given nodes in different graph topologies. After generating a rather large data set of such landscapes, they propose a graph neural network architecture which performs well on the task and can generalize to real topologies taken from electric power grids.

**Strengths:**

1. The presentation of a full pipeline for perturbation prediction is impressive and valuable. Overall, I think the paper is very sound as is, but could be improved by addressing some of the questions and weaknesses discussed below.


2. The concrete products of this study in the form of the dataset of stability landscapes and the baseline GNN encoder are useful and will help the community study these synchrony phenomena in the future.

**Weaknesses:**

1. The authors emphasize the need for stability landscapes holistically so that “safe and unsafe” regions in perturbation space can be predicted. I think in theory this is a very good idea. However, I notice that–at least for the case of the inertial Kuramoto model studied here–these landscapes have similar coarse structure: uniform zero vs full “eye structure” vs top eyelid vs bottom eyelid. I think it would be worthwhile for the authors to justify why predicting the full pointwise array of landscape probabilities is valuable compared to just predicting these coarse categories. Would it be possible to decode these categories from their GNN? This is not a weakness per se, but I think addressing this categorical structure in their data could help solidify the problem setting. For instance, we see some interesting stripes in the lower part of Fig 1a/b. Could the authors explain why predicting this fine-grained structure is interesting and maybe in parallel discuss if coarser categories are worth predicting too?


2. I was a little confused by the nature of the second downstream task which “[predicts] the volume of the basin stability, a metric derived from the heatmaps.” I believe these are the results shown in Table 3, but it is not immediately clear from the text. Could the authors clarify what is the nature of this task, even if the details are in the supplement? It also seems to connect to my first point about predicting qualitative structure as opposed to fine-grained detail.


3. I think that the focus on second-order Kuramoto is justified, and it is not worth exploring other models in extensive detail. However, I think it could be worthwhile to discuss extensions to other dynamics in the concluding sections. On its face, it seems like the method would work fine on other systems, but what challenges, if any, do the authors foresee, for example, if the landscapes or perturbations (maybe with higher order derivatives) become more complex?

**Questions:**

To summarize questions implicit in the weaknesses mentioned above:

1. What is the value in predicting fine-grained vs coarse landscape structure? Can you discrete decode landscape categories?

2. Can you clarify the nature of the the second downstream task which “[predicts] the volume of the basin stability, a metric derived from the heatmaps”?

3. Can you discuss extensions to other oscillator models and perturbations? Off the top of my head, maybe delay models, noise, higher-order interactions? (You don't have to discuss these examples specifically, but it would be good in general to read about challenges and opportunities in extending beyond the current case.

---

> ### Author Response · Authors · 2025-11-21
> **Reply to Reviewer yScr**
>
> > What is the value in predicting fine-grained vs coarse landscape structure? Can you discrete decode landscape categories?
>
> Fine‑grained structure matters for tasks where tiny perturbation changes flip outcomes, e.g. riddled or filamentary basins (<https://doi.org/10.1063/1.5012134>). Capturing these thin unstable corridors enables differentiating nodes that are globally robust from those with hidden, narrowly vulnerable directions. A promising future extension is to extract coarse landscape “types” (e.g. fully stable, broad mixed regions, riddled/filamentary) from GNN embeddings to triage where expensive high‑fidelity simulation is necessary and where it can be skipped.
>
> One challenge is that a node’s regime (mostly stable, large mixed stable/unstable regions, or dominated by fine filaments as in Fig. 1a/b) is not known a priori. Future work could investigate whether a classifier derived from the learned representations could act as a selector: (i) flag nodes requiring dense perturbation sampling; (ii) confirm nodes with no unstable region; (iii) identify candidates for riddled‑basin analysis.
>
> We added another down-stream task: Contingency screening which would not be possible without the heatmaps. (See Reply to all Reviewers: Adding another down-stream task based on landscapes: Contingency screening)
>
> > Q2: Can you clarify the nature of the the second downstream task which “\[predicts\] the volume of the basin stability, a metric derived from the heatmaps”?
>
> Clarification: We follow the wording of <https://www.nature.com/articles/nphys2516>, where “the volume of a state’s basin of attraction” quantifies the likelihood of reaching that state. The scalar SNBS is exactly this normalized basin volume. Our heatmap prediction makes the scalar directly derivable (by averaging stability over pixels), showing that landscape learning subsumes scalar SNBS prediction.
>
> > Q3: Can you discuss extensions to other oscillator models and perturbations?
>
> We expect the framework to transfer to other dynamical systems given sufficient high‑resolution training data. See Reply to all Reviewers: Generalizability and potential in other fields. More complex dynamics may increase data generation cost, but we see no conceptual obstacle to applying the same landscape formulation beyond Kuramoto models. We add this to the Conclusion section as outlook.

---

### Official Review · Reviewer_YmM5 · 2025-10-31

**Soundness:** 3
**Presentation:** 2
**Contribution:** 3
**Rating:** 4
**Confidence:** 3

**Summary:**

The paper introduces Stability Landscapes as an upstream task for node-level robustness in network synchronization, releasing two large datasets (10k graphs each at 20 and 100 nodes) generated from a conceptual power-grid oscillator model to derive many downstream stability indices.

**Strengths:**

- **Novel problem formulation**: Rather than predicting a single stability index, the paper proposes predicting a full landscape of stability across the perturbation space. This is conceptually interesting and could inspire related tasks such as node-level OOD detection or uncertainty quantification in stability analysis.
- **Substantial computational investment**: The datasets required approximately 500,000 CPU hours to generate, representing a significant resource contribution that could benefit the broader community.
- **Cross-distribution evaluation**: The evaluation includes both in-distribution performance and out-of-distribution generalization across graph sizes and to realistic power grid topologies, which is commendable.

**Weaknesses:**

- **Unclear practical advantage over scalar prediction**: The paper fails to demonstrate concrete scenarios where landscape prediction outperforms scalar SNBS prediction. Table 3 shows that deriving SNBS from predicted landscapes actually performs worse than direct SNBS prediction. Without clear use cases where the spatial structure of landscapes provides actionable insights beyond a single stability score, the motivation for this more complex task remains unconvincing. It is unlikely that practitioners would inspect heatmaps for each node qualitatively.
- **Overly restrictive experimental scope limits generalizability**: The evaluation is confined to a single oscillator model with fixed parameters, binary node features, and a specific perturbation and threshold. No parameter analysis is provided. This rigid setting raises serious questions about whether the approach generalizes to other coupled oscillator models, different parameter regimes, or other classes of networked dynamical systems. The claimed contribution to "complex networked systems across biology, neuroscience, and power grids" is not substantiated.
- **Methodological ambiguity in downsampling procedure**: The downsampling from 10,000 random perturbations per node to a 20×20 grid is poorly explained. Since the original perturbations form "a point cloud that differs from node to node in both location and density" (line 190), how does the discretization ensure consistent spatial correspondence? downsampled cells at the same grid index (m,n) may still correspond to different regions of the perturbation space across nodes.
- **Inadequate loss function and evaluation metrics for image prediction**: For image reconstruction tasks, L1 loss is more commonly employed than L2/MSE due to its robustness to outliers. More critically, evaluating heatmap quality solely with R² is insufficient. Standard image quality metrics such as PSNR, SSIM, or perceptual similarity should be reported to properly assess reconstruction fidelity.
- **Missing ablation study on heatmap resolution**: The choice of 20×20 discretization appears arbitrary and is not justified. The paper lacks any ablation study examining how performance varies with heatmap resolution (e.g., 10×10, 30×30, 40×40). This is essential to validate that the task formulation is robust, useful and to understand the trade-off between different granularity.
- **Insufficient baseline comparisons on realistic topologies**: Tables 4-5 show transfer to real power grids, but no comparison to alternative methods is provided. Given the claimed computational advantage over Monte Carlo sampling, the authors should benchmark against cheaper approximation methods or classical stability estimation techniques on these realistic networks. This omission is particularly concerning given the massive computational cost of dataset generation and training.

**Questions:**

See weakness section

---

> ### Author Response · Authors · 2025-11-21
> **Reply to Reviewer YmM5**
>
> > Weakness **Unclear practical advantage over scalar prediction**
>
> See Reply to all Reviewers: Adding another down-stream task based on landscapes: Contingency screening, which introduces a task that requires the full landscape and cannot be derived from a scalar.
>
> > **Overly restrictive experimental scope limits generalizability**
>
> See Reply to all Reviewers: Generalizability and potential in other fields.
>
> > **Methodological ambiguity in downsampling procedure**
> >
> > **Missing ablation study on heatmap resolution**:
>
> See Reply to all Reviewers: Grid resolution.
>
> > **Inadequate loss function and evaluation metrics for image prediction**
>
> We thank the reviewer for this helpful suggestion. See Reply to all Reviewers: New metrics for landscape comparison.
>
> > **Missing ablation study on heatmap resolution**
>
> We agree this is an interesting direction and added more results with a different grid size. Please see reply to all Reviewers: Grid resolution.
>
> We are happy to provide even more results for the camera-ready version.
>
> > **Insufficient baseline comparisons on realistic topologies**
>
> See Replies to all Reviewers: Complexity of stability landscapes and simple baselines; Introducing more advanced baseline models.

---

### Official Review · Reviewer_mAKt · 2025-11-01

**Soundness:** 2
**Presentation:** 2
**Contribution:** 1
**Rating:** 4
**Confidence:** 3

**Summary:**

This paper introduces the concept of Stability Landscapes, a upstream task for predicting the robustness of synchronization in dynamical systems on graphs. Instead of estimating a single scalar stability score per node (basin stability), the authors propose predicting full stability heatmaps that describe how node-level stability varies across perturbation conditions.

**Strengths:**

- The paper introduces stability landscapes as a new supervised learning task, generalizing beyond scalar stability measures.
- Two new datasets (10,000 graphs each) are released, generated from half a million CPU hours of Kuramoto simulations. This provides the research community with a valuable benchmark that combines topology, dynamics, and continuous-valued outputs.
- The method achieves ≈90% R² in-distribution, 66–70% under size-shift (20 - 100 nodes), and up to 86% on real-world power grid topologies, showing impressive zero-shot transfer across domains.

**Weaknesses:**

- The GNN–MLP architecture is quite simple and largely serves as a proof of concept. The innovation lies more in the dataset and task formulation than in methodological novelty.
- The work focuses exclusively on Kuramoto oscillator models. While this is a canonical setting, it limits the immediate applicability to broader classes of dynamical systems (biochemical, ecological, or chaotic dynamics).
- Although the landscapes provide richer information, the paper doesn’t explore what the learned embeddings represent physically, or how topological features influence predicted stability patterns.
- Despite claiming open release, the dataset generation process requires massive compute (∼500,000 CPU hours), making reproducibility difficult.
- Given that the encoder-decoder is trained on synthetic graphs, even though it generalizes reasonably, the robustness on heterogeneous, real-world network distributions (with different coupling constants, noise, or weighted edges) remains uncertain.

**Questions:**

See the points mentioned in the weaknesses.

- Could the Stability Landscape framework be extended to other classes of dynamical systems (chaotic oscillators or neural mass models)? How general is the proposed formulation?

- Why were simple decoders (MLPs) chosen instead of more expressive generative models (VAEs, CNNs, or diffusion-based decoders) for 2D heatmap reconstruction?

- What structural graph features most influence the predicted stability landscapes?

- How would the approach perform on much larger real grids (>1,000 nodes)? Are there computational bottlenecks or accuracy drop-offs with scale?

---

> ### Author Response · Authors · 2025-11-21
> **Reply to Reviewer mAKt**
>
> > Weakness: The work focuses exclusively on Kuramoto oscillator models. While this is a canonical setting, it limits the immediate applicability to broader classes of dynamical systems (biochemical, ecological, or chaotic dynamics).
> >
> > Weakness: Given that the encoder-decoder is trained on synthetic graphs, even though it generalizes reasonably, the robustness on heterogeneous, real-world network distributions (with different coupling constants, noise, or weighted edges) remains uncertain.
> >
> > Q: Could the Stability Landscape framework be extended to other classes of dynamical systems (chaotic oscillators or neural mass models)? How general is the proposed formulation?
>
> See Reply to all Reviewers: Generalizability and potential in other fields.
>
> > Weakness: Although the landscapes provide richer information, the paper doesn’t explore what the learned embeddings represent physically, or how topological features influence predicted stability patterns.
> >
> > Question: What structural graph features most influence the predicted stability landscapes?
>
> Studying the learned embeddings is an interesting research direction, but it is out of scope for this paper. For the influence of topological features, please see Reply to all Reviewers: Complexity of stability landscapes.
>
> > Despite claiming open release, the dataset generation process requires massive compute (∼500,000 CPU hours), making reproducibility difficult.
>
> The full unprocessed trajectory data is too large to reasonably make available (For Dataset100 it would be in the order of 100 terabytes). Our release enables reproducing stability predictions at the level of individual grids, nodes, and even perturbations, which substantially reduces the compute required for (partial) replication.
>
> > Why were simple decoders (MLPs) chosen instead of more expressive generative models (VAEs, CNNs, or diffusion-based decoders) for 2D heatmap reconstruction?
>
> As this is the first work to study graph-to-landscape prediction, we intentionally use a simple and transparent GNN–MLP baseline to isolate the difficulty of the task itself,  and provide a clear benchmark for future methods. We release all data and code to facilitate testing of more expressive decoders. More advanced decoders (VAEs and diffusion models) are promising next steps, and we view the present model as a clean baseline for such future exploration. Following the reviewers suggestions, we also introduce additional baselines using CNN decoding; see Reply to all Reviewers: Introducing more advanced baseline models.
>
> > How would the approach perform on much larger real grids (>1,000 nodes)? Are there computational bottlenecks or accuracy drop-offs with scale?
>
> The problem is to conduct probabilistic dynamical simulations of larger grids, since the computational costs grow at least quadratically. GNNs do not face this problem at all, since they consider a subgraph only (depending on number of hops). Hence, the out-of-distribution generalization is of great real interest. If ML models can be trained on small subgraphs, but still reliably predict the dynamic behavior of full-sized grids, grid operators would be allowed to analyze large number of contingencies of real-sized grids.
>
> The main bottleneck is generating probabilistic simulation data, whose cost grows at least quadratically with grid size. In contrast, GNN inference scales with local neighborhoods (fixed hop radius), so prediction remains tractable on large graphs. This makes out-of-distribution generalization across size especially relevant: training on small subgraphs while reliably predicting on full networks could enable screening large numbers of contingencies on >1,000‑node grids.

---

### Official Review · Reviewer_xjid · 2025-11-01

**Soundness:** 2
**Presentation:** 3
**Contribution:** 2
**Rating:** 4
**Confidence:** 3

**Summary:**

This paper introduces "stability landscapes" as a novel upstream task for predicting per-node stability behavior in dynamical systems on networks. Instead of predicting scalar stability indices (SNBS), the authors propose predicting full 2D stability landscapes (20×20 heatmaps) that capture spatial patterns of stability across perturbation space. The work focuses on second-order Kuramoto oscillator networks and uses GNN-based encoder-decoder architectures, achieving ~90% R² in-distribution and 66-70% under size shift.

**Strengths:**

Originality
  - Novel problem formulation: First work to predict full stability landscapes rather than scalar stability indices, representing a meaningful extension beyond prior SNBS prediction work
  - Creative task design: The graph-to-image prediction task is unique in the stability analysis domain and could inspire similar approaches in other fields
  - Theoretical contribution: Theorem 3.1 provides a clean mathematical relationship between pixel-wise MSE and downstream SNBS error, offering theoretical grounding for the approach

Quality

  - Substantial dataset effort: 500k CPU hours to generate datasets with 10k graphs each at two scales demonstrates significant computational investment
  - Multiple evaluation scenarios: Tests in-distribution, out-of-distribution (size shift), and zero-shot transfer to real power grids
  - Sound experimental methodology: Proper train/validation/test splits, multiple seeds, and statistical reporting

Clarity

  - Clear motivation: The progression from scalar to landscape prediction is well-motivated and easy to follow
  - Good mathematical exposition: Equations and notation are generally clear and consistent
  - Effective visualizations: Figure 1 effectively illustrates the concept, and landscape comparisons in Figures 4,6,7 are informative

Significance

  - Potential broader impact: Could influence how stability analysis is performed across multiple domains (power grids, neuroscience, biology)
  - Methodological advancement: Demonstrates feasibility of predicting complex spatial patterns from graph topology alone

**Weaknesses:**

Experimental Validation Gaps

  1. Missing critical baselines: No comparison to interpolation methods, PCA-based reconstruction, or modern generative models (VAEs, GANs, diffusion models) that are natural baselines for image generation tasks
  2. Limited architecture exploration: Only tests two GNN encoders with simple MLP decoders; no exploration of CNN decoders, attention
  mechanisms, or other architectures suited for image generation
  3. Inadequate evaluation metrics: Relies solely on R² which doesn't capture perceptual quality or spatial structure; missing image-specific
  metrics like SSIM, LPIPS, or FID

  Questionable Practical Utility

  1. No evidence of superior decision-making: Authors claim landscapes enable "multiple downstream tasks" but only demonstrate SNBS recovery, which often performs worse than direct SNBS prediction (Table 3: 82-89% vs 85-90%)
  2. Missing use case analysis: No concrete examples of decisions that would benefit from spatial landscape information vs scalar indices

  Technical Limitations

  1. Poor generalization: Significant performance drop from 90% to 66-70% R² under size shift, and high variance on real networks (32-86% R²) suggests limited practical reliability
  2. Arbitrary design choices: 20×20 discretization appears unmotivated; no sensitivity analysis or principled approach to grid resolution
  selection
  3. Information loss in preprocessing: Point cloud to grid conversion discards information and introduces artifacts, but this trade-off is
  not analyzed

  Scope Limitations

  1. Single dynamical model: Restricted to second-order Kuramoto oscillators; generalization to other dynamical systems unknown
  2. Limited network diversity: Despite claims of broad applicability, only tested on power grid topologies
  3. Computational efficiency unclear: Authors claim speed advantages but provide no timing comparisons or scalability analysis

**Questions:**

1. Computational efficiency: What are the actual runtime and memory comparisons between landscape prediction vs direct simulation vs direct SNBS prediction? This is crucial for practical adoption claims.
  2. Grid resolution sensitivity: How does performance vary with different grid resolutions (10×10, 30×30, etc.)? Why was 20×20 chosen
  specifically?
  3. Baseline comparisons: How does the approach compare to:
    - Simple interpolation of scattered simulation points?
    - PCA reconstruction from a subset of simulations?
    - Modern generative models trained on the same data?
  4. Practical utility demonstration: Can the authors provide specific examples where landscape spatial information leads to better
  engineering decisions than scalar indices?
  5. Failure mode analysis: When and why does the method fail? What network characteristics predict poor performance?
  6. Architecture justification: Why use simple MLPs instead of CNNs or other architectures designed for image generation? What's the
  performance gap?
  7. Real-world validation: Have actual power grid operators evaluated whether these landscapes provide useful insights for grid planning and operation?

---

> ### Author Response · Authors · 2025-11-21
> **Reply to Reviewer xjid**
>
> > Q1: Computational efficiency: What are the actual runtime and memory comparisons between landscape prediction vs direct simulation vs direct SNBS prediction? This is crucial for practical adoption claims.
>
> Conducting the dynamical simulations requires high amounts of CPU hours. The computational costs increase with the grid size (scales at least quadratically), but for 10,000 grids of size 100 with 10,000 perturbations per node, the cost is roughly 450,000 CPU hours. Per allocated CPU, 4-8GB of memory are needed, so on average 45 CPU hours to generate the landscapes for a single grid, however, for unstable grids the simulation time can increase to up to a week.
>
> Inference of either a scalar SNBS value or a full landscape for a node takes <1 second wall time and negligible additional memory beyond loading the trained weights, making prediction essentially free compared to simulation. For predicting the landscapes, larger models are used (DBGNN roughly 3.6 million parameters, TAG roughly 18 million parameters), whereas for the simple scalar predictions, the model had less than 1 million parameters.
>
> > Q2: Grid resolution sensitivity: How does performance vary with different grid resolutions (10×10, 30×30, etc.)? Why was 20×20 chosen specifically?
>
> See Reply to all Reviewers: Grid resolution.
>
> > Q6: Architecture justification: Why use simple MLPs instead of CNNs or other architectures designed for image generation? What's the performance gap?
> >
> > W2: Limited architecture exploration
>
> See Reply to all Reviewers: Introducing more advanced models.
>
> > Q3: Baseline comparisons: How does the approach compare to: - Simple interpolation of scattered simulation points? - PCA reconstruction from a subset of simulations? - Modern generative models trained on the same data?
> >
> > W1: Missing critical baselines
>
> The idea to reconstruct the landscape from limited simulation data is intriguing. Does the reviewer have a reference in mind that does something like this?
>
> One issue in our setting is that even limited simulation data is expensive. For a relatively unstable grid, obtaining 100 samples per node, so 1% of the full set, could take more than 1.5 CPU hours.
>
> Simple interpolation of sparse samples fails because many basins exhibit sharp, filamentary, and riddled boundaries; upsampling introduces artifacts and cannot recover fine unstable filaments visible in Fig. 1b. These structures are not smoothly varying and violate assumptions behind standard spatial interpolation. See Reply to all Reviewers: Complexity of stability landscapes and simple baselines. But PCA based reconstruction appears more plausible. If the reviewer can provide a reference or details for a concrete method we will try to incorporate this into the final manuscript.
>
> > Q4: Practical utility demonstration: Can the authors provide specific examples where landscape spatial information leads to better engineering decisions than scalar indices?
>
> Landscapes enable tasks not possible from a single SNBS scalar. For example estimating the direction and minimal magnitude of a perturbation that destabilizes a node with high probability (see Adding another down-stream task based on landscapes: Contingency screening); This helps to prioritize contingencies by identifying narrow unstable “corridors” vs broad vulnerable regions.
>
> > Q5: Failure mode analysis: When and why does the method fail? What network characteristics predict poor performance?
>
> See Reply to all Reviewers: Complexity of stability landscapes and simple baselines.
>
> > Q7: Real-world validation: Have actual power grid operators evaluated whether these landscapes provide useful insights for grid planning and operation?
> >
> > Missing use case analysis:
>
> Our contribution establishes technical feasibility of predicting stability landscapes. Similar probabilistic landscape concepts are already used (e.g., probabilistic short‑circuit landscapes: <https://ieeexplore.ieee.org/document/9494855>). While our dataset is not sized or parameterized for direct operator deployment, it paves the way for follow‑up studies with operational data.
>
> The second added down-stream task highlights the potential for contingency screening (See reply to all Reviewers: Adding another down-stream task based on landscapes: Contingency screening)
>
> > W3: Inadequate evaluation metrics:
>
> We thank the reviewer for this helpful suggestion.
>
> See Reply to all Reviewers: New metrics for landscape comparison

---

### Author Response · Authors · 2025-11-21
**Reply to all reviewers 1**

We thank the reviewers for their constructive feedback. Across reviews, strengths clustered into four areas:

* Originality: Novel problem formulation (Reviewer YmM5); creative task design (Reviewer xjid)
* Quality: Sound experimental methodology, multiple evaluation scenarios (Reviewer xjid); commendable cross-distribution evaluation (Reviewer YmM5); impressive zero-shot transfer across domains (Reviewer mAKt)
* Clarity: Clear motivation, good mathematical exposition, effective visualizations (Reviewer xjid); the paper is very sound as is (Reviewer yScr)
* Significance: a valuable benchmark (Reviewer mAKt); could benefit the broader community (Reviewer YmM5); an impressive and valuable end‑to‑end pipeline (Reviewer yScr)

Below, we first address the points raised by multiple reviewers, followed by detailed individual responses.

## New metrics for landscape comparison

We thank the reviewers for the great idea and included the following metrics: SSIM and LPIPS for comparing the landscapes (images).

We added two standard image-quality metrics, **SSIM** and **LPIPS**, to evaluate the predicted heatmaps.  These metrics have been added to the revised manuscript as well.

For convenience, we also include the main result here.

**Performance on heatmap prediction measured by SSIM (%)**

| Model | tr20ev20 | tr100ev100 | tr20ev100 |
|-------|----------|------------|-----------|
| TAG-MLP | 81\.32 ± 0.55 | 81\.61 ± 0.15 | 71\.50 ± 0.28 |
| DBGNN-MLP | 80\.63 ± 0.21 | 82\.87 ± 0.79 | 74\.03 ± 0.46 |
| TAG-CNN | 76\.47 ± 0.37 | 70\.51 ± 0.48 | 67\.46 ± 0.56 |
| **DBGNN-CNN** | **84\.24 ± 0.27** | **85\.89 ± 0.21** | **78\.50 ± 0.19** |

## Introducing more advanced models

Following the suggestion of multiple reviewers, we added more complex baseline models. By using CNN decoders the performance can be improved in comparison to decoders with MLPs. The new results are included in the paper. Since we publish all code and data, other groups can easily test other architectures. We are also happy to add more complex architectures to the camera-ready version. Using a CNN decoder improves the in-distribution performance by roughly 3 % in terms of SSIM. For results, see also: Reply to all reviewers: New metrics for landscape comparison.

## Adding another down-stream task based on landscapes: Contingency screening

Understanding geometric features of the space from which the system can recover to normal operation is a central theme in engineering and mechanics \[<https://ieeexplore.ieee.org/abstract/document/8725530>\], but also eco systems and other networked systems \[<https://www.nature.com/articles/s41598-020-68805-6>\], with ongoing research \[<https://journals.aps.org/prl/abstract/10.1103/PhysRevLett.127.194101>\]. A typical object of study is the size of the minimal shock that can destabilize the system \[<https://www.nature.com/articles/s41598-020-68805-6>\]. The probabilistic version of this concept, the radius at which the probability to desynchronize crosses some threshold, is studied as the linear size of the basin in \[<https://pubs.aip.org/aip/cha/article/27/10/103109/151447/The-size-of-the-sync-basin-revisited>\]. The radial projection of our stability landscape is exactly what is estimated directly numerically in this paper.

The linear size of the basin thus provides an immediate downstream task. From a power grid perspective, a further refinement is more interesting: Finding not just the radius, but also the precise direction of perturbation at which we first see a high likelihood of failure. In the power grid operator setting this provides a type of contingency screening: As not all possible contingencies can be studied, the basin landscape allows identifying a set of minimal perturbation regions that have a high likelihood of destabilizing the grid. These can then be singled out for further in-depth analysis, to understand why the system is prone to failure here.

The new task is in a new subsection in the paper: Identification of critical contingencies.

---

> ### Author Response · Authors · 2025-11-21
> **Reply to all Reviewers 2**
>
> ## Generalizability and potential in other fields
>
> Our results demonstrate the general feasibility of learning stability landscapes from graph-structured dynamical systems, and we hope this encourages domain experts in other areas to explore the approach. As noted in the section on down-stream tasks, the task we introduce is not specific to Kuramoto or power grid settings, minimal shocks are considered in a wide range of settings. Beyond Kuramoto oscillators, GNNs have already been shown to predict probabilistic dynamics of more realistic inverter‑rich power grids with heterogeneous node and edge features <http://arxiv.org/abs/2406.08917>. Given sufficiently high‑resolution training data (dense perturbation sampling), we see no conceptual barrier to extending landscape prediction to other oscillator models (chaotic, neural mass), or to biochemical and ecological network dynamics; the main practical constraint is designing good datasets and the cost of generating simulation data.
>
> The referenced work <http://arxiv.org/abs/2406.08917> used only 1,000 perturbations per node (vs. 10,000 samples in our dataset), which is too sparse to produce high‑fidelity landscapes. By releasing our higher‑resolution data, we aim to motivate the creation of richer benchmarks that enable reliable landscape learning across domains.
>
> ## Complexity of stability landscapes and simple baselines
>
> The probabilistic dynamics underlying the stability landscapes are highly non‑linear, making them difficult to approximate with simple heuristics. Identifying stability patterns remains an open research problem in network science: while correlations between standard network measures and stability can be observed, they are not causal and fail to generalize across topologies (<https://arxiv.org/abs/2402.17500>).
>
> There is no known theoretical shortcut to directly predict single‑node basin stability (SNBS). Prior work applying regression or MLP models to handcrafted network features yields clearly inferior performance to GNNs, so we did not include those weaker baselines (<https://doi.org/10.1063/5.0160915>).
>
> If the reviewers have other baselines in mind, we are happy to include them.
>
> ## Grid resolution
>
> We selected a 20×20 grid (400 cells) as the highest practical resolution with acceptable sampling noise: 10,000 perturbations / 400 cells ≈ 25 samples per pixel. Finer grids would introduce sparsity and higher variance per cell; coarser grids would lose filamentary structure. All raw perturbation data are released, enabling recomputation at alternative resolutions. To show the general feasibility, we provide results of the TAG-MLP when using 10x10 grids (100 cells per grid). Overall, the results show that the approach is feasible with different resolution. We are happy to add more resolutions to the camera-ready version.
>
> ## Revised manuscript uploaded
>
> We have uploaded a revised manuscript that includes new metrics, new baseline models with CNN decoders, a new downstream task, and an analysis of grid resolution (see Appendix). We will continue to improve the paper until the camera-ready version, focusing on enhancing the flow and transitions to reflect these changes.

---

### Author Response · Authors · 2025-12-02
**Summary of Rebuttal Outcomes**

Dear ACs, SACs, and PCs,

We thank your time and effort in managing our submission and overseeing the review, especially in light of the recent incident and the added workload.

We also thank the reviewers for their thoughtful feedback, which substantially improved our paper.

---

In the rebuttal, we addressed the main concerns raised across all four reviews. Key improvements include:


**1\. Expanded evaluation metrics and decoder architectures (Reviewers xjid, mAKt, YmM5)**

We incorporated SSIM and LPIPS as standard image-quality metrics alongside R², and introduced advanced decoders—CNN-based and Transformer-based—on top of GNN encoders. These enhancements improved in-distribution SSIM scores by approximately 3 percentage points over MLP decoders (**\*\*Tables 2-5, Appendix B.5\*\***). The use of robust metrics and more complex models strengthened our experimental setup, while the core results and storyline of the paper remained unchanged.

| Model | tr20ev20 | tr100ev100 | tr20ev100 |
|-------|----------|------------|-----------|
| TAG-MLP | 81\.32 ± 0.55 | 81\.61 ± 0.15 | 71\.50 ± 0.28 |
| DBGNN-MLP | 80\.63 ± 0.21 | 82\.87 ± 0.79 | 74\.03 ± 0.46 |
| TAG-CNN | 76\.47 ± 0.37 | 70\.51 ± 0.48 | 67\.46 ± 0.56 |
| DBGNN-CNN | 84\.24 ± 0.27 | **85\.89 ± 0.21** | **78\.50 ± 0.19** |
| TAG-ViT | **84\.41 ± 1.03** | 82\.25 ± 3.97 | 75\.47 ± 1.62 |

**2\. Introduced a new downstream task reflecting grid operator needs (Reviewers YmM5, xjid, yScr)**

We added Section 4.4, “Identification of critical contingencies,” which leverages the full landscape to determine the direction and minimal size of destabilizing perturbations—information not accessible from a single scalar (**\*\*Figure 5, Appendix B.6\*\***). This demonstrates the practical advantage of predicting image landscapes.

**3\. Clarified discretization, grid resolution, and robustness (Reviewers xjid, YmM5)**

We detailed the downsampling process from perturbation point clouds to 2D grids, explained how spatial correspondence is maintained, and justified the choice of a 20×20 grid (balancing detail and sampling noise). Additional results for 10×10 grids confirm that our approach and conclusions are robust to resolution changes (**\*\*Appendix B.7\*\***).

**4\. Addressed baselines, landscape complexity, generalizability, and computational efficiency (all Reviewers)**

We explained the limitations of simple interpolation and heuristic baselines for riddled/filamentary basins, and discussed generative reconstruction baselines as promising future work.

We clarified that our approach is not restricted to Kuramoto oscillators, but is applicable to other dynamical systems with high-resolution training data. Connections to existing GNN work on realistic inverter-rich power grids were made, positioning our dataset as a high-resolution benchmark for future extensions.

Simulation cost versus inference time (<1 second per node) was quantified, and we highlighted that GNN inference scales efficiently to large grids. Our data release strategy—providing processed landscapes and perturbation outcomes rather than raw trajectory data (\~100 TB)—balances practicality and reproducibility.

---

### Meta-Review · Area_Chair_bbxN · 2026-01-06

**Summary:**

Reviewers recognized the novelty of the task and the dataset effort, but raised concerns about the limited methodological novelty and weak empirical support for broad generalization claims. These concerns informed the recommendation.

**Reviewer Concerns:**

Several presentation and evaluation issues were addressed in the rebuttal (additional metrics, architectures, and clarifications). However, key concerns remain: the methodological novelty is limited, generalization claims remain speculative, and comparisons to alternative baselines are missing.

**Reviewer Scores:**

Reviewer scores would likely remain unchanged. While one reviewer was positive, multiple reviewers remained below the acceptance threshold and would likely maintain their assessments.

---

### Decision · Program_Chairs · 2026-01-26

Reject